**Cite this article:** da Silva PFL, Schumacher B. 2019 DNA damage responses in ageing. *Open Biol.* **9**: 190168.

**Subject Area:**
cellular biology/developmental biology/genetics

ageing, DNA repair, cellular senescence, longevity, DNA damage response

**Author for correspondence:**
Björn Schumacher
e-mail: bjoern.schumacher@uni-koeln.de

# DNA damage responses in ageing

Paulo F. L. da Silva[1,2] and Björn Schumacher[1,2]

[1]Institute for Genome Stability in Ageing and Disease, Medical Faculty, and [2]Cologne Excellence Cluster for Cellular Stress Responses in Ageing-Associated Diseases (CECAD), Center for Molecular Medicine Cologne (CMMC), University of Cologne, Joseph-Stelzmann-Strasse 26, 50931 Cologne, Germany

BS, 0000-0001-6097-5238

Ageing appears to be a nearly universal feature of life, ranging from unicellular microorganisms to humans. Longevity depends on the maintenance of cellular functionality, and an organism's ability to respond to stress has been linked to functional maintenance and longevity. Stress response pathways might indeed become therapeutic targets of therapies aimed at extending the healthy lifespan. Various progeroid syndromes have been linked to genome instability, indicating an important causal role of DNA damage accumulation in the ageing process and the development of age-related pathologies. Recently, non-cell-autonomous mechanisms including the systemic consequences of cellular senescence have been implicated in regulating organismal ageing. We discuss here the role of cellular and systemic mechanisms of ageing and their role in ageing-associated diseases.

## 1. Introduction

Ageing can be defined as a state of progressive functional decline accompanied by an exponential increase in mortality (the Gompertz law [1–3]). Despite being widespread among almost all multicellular organisms [4,5], there are exceptions. The existence of species without an observable time-dependent functional decline and increase in mortality, termed 'negligible senescence' [6–8], suggests that the ageing process is not an entirely ubiquitous, inevitable one, hence raising the important questions of 'why does it happen?' and 'how can it be so variable?'

In the wild, extrinsic factors are the ones mostly leading to mortality: animals tend not to grow very old and, as a result, the power of natural selection declines over time. Natural selection is, therefore, predicted to only have a weak influence on the process of senescence, making the existence of genes that actively promote ageing very unlikely [9]. Instead, according to the 'mutation accumulation' theory, this lack of selective effects in later life stages allows the accumulation of alleles with late, unselected effects over several generations [9]. Alternatively, the 'antagonistic pleiotropy' theory proposes a major contribution of pleiotropic genes—genes selected to maintain fitness during early life but with unselected deleterious effects later, after the organism's reproductive period—in the development of age-related phenotypes [4]. Finally, the 'disposable soma' theory states that, because of resource scarcity, organisms evolved mechanisms to optimally allocate metabolic resources into reproduction at the expense of somatic maintenance. Proper somatic maintenance is only required to ensure that an organism reaches reproductive maturity; therefore, it can be beneficial not to invest resources into somatic repair and maintenance even if that will lead to damage accumulation over time, ultimately driving the ageing process [5,10]. Both the 'antagonistic pleiotropy' and 'disposable soma' theories are based on the idea of a cellular 'trade-off'—a compromise, where mechanisms that are at first advantageous bring detrimental consequences later on.

In this review, we primarily focus on the role of DNA damage accumulation in pathology and in the ageing process. We start by highlighting evolutionary trade-offs between somatic maintenance and reproduction and how these can

be tightly connected to an organism's environment; we then move on to the role of DNA repair pathways in enforcing these trade-offs at the cellular and organismal level. Finally, we give special attention to non-cell-autonomous DNA damage responses (DDRs), which promote tissue dysfunction and compensatory responses with the aim of re-establishing tissue homeostasis, as the study of these will surely facilitate the identification of the mechanisms underlying the systemic effects of DNA damage in the future.

## 2. Influence of the environment on lifespan and ageing

Seeing that organisms inhabit highly variable environments, it is not surprising that there is a vast range of highly specialized traits and responses that promote survival and/or optimal reproduction for those specific environments and circumstances. In the nematode *Caenorhabditis elegans*, some of these environmental responses are very well characterized. Temperature is a major influencing factor in the nematode's lifespan: hermaphrodite worms have a half-life of about 13 days at 20°C but their lifespan can be modulated by the environment's temperature [11]. A decrease in temperature to 16°C increases the worms' half-life to approximately 20 days, while an increase in temperature to 25°C has the opposite effect, decreasing half-life to approximately 8 days [11].

In addition to temperature, food availability is also a major modulator of the worms' lifespan. When faced with starvation, *C. elegans* can enter diapause—a physiological state of dormancy and developmental delay, with halted feeding and reproduction [12–14]. Depending on the developmental stage at which the worms face starvation, distinct diapause states can be established. Dauer arrest, the most well-studied diapause state, is established when worms are starved at the L2 larval stage. Dauer worms undergo specific anatomical and metabolic modifications and are surprisingly resistant to different environmental stressors when compared with non-dauer worms [14–16]. Importantly, worms have been reported to survive up to several months in this stage but are still able to resume development, reach adulthood and display normal adult lifespan and reproduction when faced again with ideal conditions [14]. A distinct diapause state—adult reproductive diapause (ARD)—is established under conditions of starvation and high larval density shortly after the transition from the L4 larval stage into young adulthood [17]. Unlike dauer, this state is not associated with major anatomical changes and, while in this state, worms show some signs of tissue and cellular ageing, including atrophy of the intestine and germline degradation [17]. Remarkably, shortly after exiting ARD, worms display normal adult morphology (including a repopulated germline and functional intestine) and lifespan [17]. This rejuvenation process becomes even more extraordinary because it takes place in adult worms, in which all somatic cells are postmitotic, strongly hinting towards the existence of signalling pathways promoting tissue functionality or, in this case, rejuvenation in a systemic way following stressful conditions. The mediators involved in this rejuvenation process are still unknown; however,

reactivation of RNA metabolism appears to be a requirement for somatic restoration post-ARD to occur [18].

The impact of food availability in stress responses and longevity is not restricted to *C. elegans*. The effects of caloric restriction (CR) (reduction in caloric intake without malnutrition) in slowing the ageing process and promoting health have been extensively described in animal models and recent studies have suggested that this might translate even to human health maintenance [19–21]. CR is hypothesized to trigger an evolutionary conserved adaptive response for periods of food scarcity responsible for shifting an organism's energy resources from growth and reproduction to somatic maintenance [22,23]. The trade-off becomes particularly apparent in *C. elegans*, where extraordinarily long-lived dauer larvae maintain somatic function until they resume offspring generation once food becomes available and are then subject to age-related somatic decline. According to the disposable soma theory, it is not surprising to observe a correlation between longevity and the amount of resources applied to somatic maintenance and repair. This is well supported by classical studies reporting a correlation between DNA repair capacity and mammalian lifespan [24,25].

## 3. The DNA damage response

One important aspect of the ageing process is the accumulation of DNA damage through time [26,27]. While containing the entire genetic information (except for mitochondria-encoded genes), the nuclear genome is constantly threatened by genotoxic insults, with an estimated frequency of the order of tens of thousands per day [28]. These hazards can arise from exogenous or endogenous sources. Exogenous sources are, to some extent, avoidable; these include ultraviolet (UV) and ionizing radiation and a variety of genotoxic chemicals. Endogenous sources, on the other hand, are unavoidable as they include metabolic by-products, such as reactive oxygen species (ROS), and spontaneous chemical reactions that target DNA molecules (including alkylation and hydrolysis of DNA chemical bonds) [28,29]. The lesion type inflicted on the DNA greatly depends on the source of the damage. Lesions caused by endogenous sources tend to arise stochastically at a higher rate. Single-strand breaks (SSBs) constitute the majority of DNA lesions, as they can arise from base hydrolysis and oxidative damage [30]. Stochastic errors during DNA replication occur at a low rate but may lead to single-nucleotide substitutions and ROS cause oxidative DNA lesions such as 8-oxoguanine [31]. Lesions caused by exogenous sources can be mutagenic and also highly cytotoxic. For instance, exposure to UV radiation leads to helix-distorting lesions such as 6–4 photoproducts [32] and, most predominantly, cyclobutane pyrimidine dimers [33]; chemotherapeutic interventions can also induce interstrand cross-links (ICLs) and double-strand breaks (DSBs) [34,35] (figure 1).

DNA damage can have distinctive consequences for cells. Persistent nucleotide substitutions, due to erroneous repair followed by misreplication, lead to the accumulation of permanent mutations and chromosomal aberrations, which increase the risk of cancer development [36]. By contrast, bulky types of DNA lesions can block transcription and replication, triggering the arrest of the normal cell cycle, ultimately leading to cell senescence or cell death, both

royalsocietypublishing.org/journal/rsob    Open Biol. 9: 190168

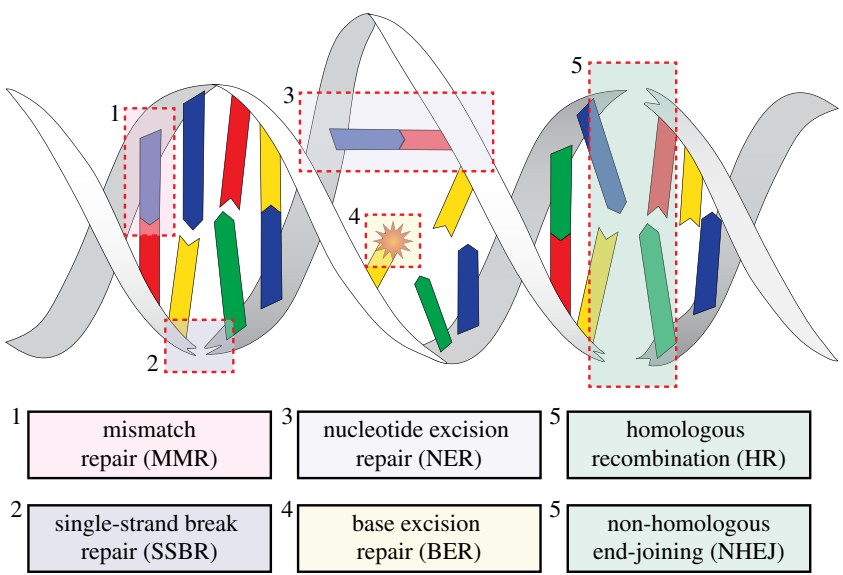

| 1 | mismatch repair (MMR) | 3 | nucleotide excision repair (NER) | 5 | homologous recombination (HR) |
|---|---|---|---|---|---|
| 2 | single-strand break repair (SSBR) | 4 | base excision repair (BER) | 5 | non-homologous end-joining (NHEJ) |

**Figure 1.** Different types of DNA lesions and corresponding DNA repair systems. Distinct DNA lesions have distinct consequences for a cell. Nucleotide substitutions followed by misreplication lead to accumulation of mutations and chromosomal aberrations, increasing the risk of cancer development. By contrast, bulkier lesions can also block replication and transcription, leading to cell-cycle arrest and, possibly, cell senescence or apoptosis. To avoid this, cells have evolved complex, highly conserved DNA repair systems capable of responding to specific types of lesions. Base mispairs (1) and short deletions/insertions are repaired by mismatch repair (MMR). Single-strand breaks (2) are repaired by complex SBBR cascades. Helix-distorting lesions, such as cyclobutane pyrimidine dimers (3), are repaired by the nucleotide excision repair (NER) pathway. Oxidative lesions and small alkylation products (4) are repaired by base excision repair (BER). Highly cytotoxic double-strand breaks (5) are either repaired by the efficient but error-prone non-homologous end-joining (NHEJ) pathway or by the precise homologous recombination (HR) pathway.

states preventing the cell from transforming into tumour cells but ultimately contributing to ageing [36]. Nuclear DNA requires constant maintenance to be kept intact and error-free in order to avoid the aforementioned consequences. For this, cells evolved intricate, evolutionarily highly conserved machineries mediating cellular responses to DNA damage—termed the 'DDR'. These highly complex systems include not only several repair pathways specific for different types of lesion but also distinct signalling cascades of damage sensors, signal boosters and effectors responsible for deciding the cell's fate. This system has two immediate goals: (i) arrest the cell cycle to prevent the propagation of corrupted genetic information, while providing time to repair the damage, and (ii) actually coordinate the repair of the DNA lesion. Depending on the success of these previous steps, the cell's fate is then decided: after lesions are successfully repaired, the DDR signalling ceases, cells survive and return to their original state; however, impossible to repair lesions trigger a persistent DDR signalling which can then engender cellular senescence or apoptosis [37,38]. Given the harmful consequences of irreparable DNA damage, it is not surprising that defects in DNA repair pathways are associated with severe human pathological conditions.

## 4. Genome instability syndromes

Human genome instability syndromes support the link between genome stability and human health, particularly premature ageing and cancer. These syndromes are typically characterized by chromosomal instability and hypersensitivity to DNA-damaging agents, thus increasing cancer predisposition and exacerbating the progressive degeneration of specific tissues [36,39–41]. Owing to the large variety of DNA lesions, cells evolved specialized, lesion-specific repair

systems. Defects in these repair systems can, however, have highly distinct functional consequences.

The most common DNA lesions, SSBs, are repaired by complex single-strand break repair (SBBR) signalling cascades initiated when the sensor protein poly(ADP-ribose) polymerase 1 (PARP1) detects and binds to SBBs [30,42]. This is followed by DNA end-processing, gap filling and ligation. Deficiencies in different factors involved in SBBR result in severe neurodegenerative phenotypes. Patients with defects in the DNA end-processing factors aprataxin, tyrosyl-DNA-phosphodiesterase 1 and polynucleotide kinase/phosphatase develop different types of cerebellar ataxia and microcephaly with sensitivity to genotoxic agents [43–50].

Oxidative and helix-distorting lesions are common types of DNA damage and are mostly repaired by three major excision repair pathways. In one of these pathways, base excision repair (BER), a DNA glycosylase recognizes and excises small chemical modifications such as oxidative lesions and small alkylation products and SSBs triggering a downstream repair signalling cascade [51,52]. BER is the main mechanism countering the deleterious effects caused by ROS, often regarded as drivers of the ageing process. It is possible that BER dysfunction plays a significant role in age-related phenotypes as it has been shown that several tissues in old mice display reduced BER capacity [53]. Importantly, age-associated neurodegenerative diseases such as Alzheimer's and Parkinson's diseases have also been linked to increased oxidative DNA damage [51,54,55] and BER has been shown to be impaired in the brains of sporadic Alzheimer's disease patients [56].

A second pathway, mismatch repair (MMR), corrects base mispairs and short deletion/insertion loops originated from replication errors, thus becoming a critical system ensuring maintenance of genome stability following DNA replication [57].

The last of the major excision repair pathways, nucleotide excision repair (NER), removes helix-distorting DNA lesions

in four consecutive steps: (i) lesion recognition; (ii) DNA unwinding; (iii) damage excision; and (iv) DNA synthesis and ligation. Two distinct lesion-recognition systems initiate the same downstream machinery, allowing the differentiation of two NER sub-pathways: (i) transcription-coupled NER (TC-NER), activated by RNA polymerase II stalling during transcription by chromatin remodelling proteins Cockayne's syndrome protein A (CSA) and B (CSB), and (ii) global genome NER (GG-NER), initiated by the UV-damaged DNA-binding protein (UV-DDB) complex and xeroderma pigmentosum group C (XPC) protein, which scan the entire genome, independently of transcription [58].

Syndromes caused by inherited defects in the NER machinery are rare and are surprisingly heterogeneous in terms of symptoms, despite a common feature of hypersensitivity to sunlight. Defects in the NER enzyme genes *XPA*, *XPB*, *XPC*, *XPD*, *XPE*, *XPF* and *XPG* can cause xeroderma pigmentosum owing to a defective GG-NER. GG-NER defects lead to the accumulation of lesions across the entire genome and, therefore, it is not surprising to observe that patients with xeroderma pigmentosum, in addition to sun-induced pigmentation abnormalities, also display a dramatically increased risk of skin cancer and internal tumours [59]. By contrast, defects in TC-NER do not inherently lead to an increased mutational load; instead, cells remain in a state of blocked transcription that ultimately leads to apoptosis. Mutations in the aforementioned lesion-recognition genes *CSA* and *CSB* can cause Cockayne's syndrome (CS). Patients with CS display a range of symptoms associated with accelerated ageing, including growth/development impairment, severe neurological defects, hearing loss, cataracts and cachexia (for an extensive review of the clinical features, see [60]), reflecting the systemic consequences of the elimination of cells with low levels of transcription-blocking DNA lesions. In addition, specific point mutations in the NER helicase genes *XPB* and *XPD* can also cause trichothiodystrophy, a severe progeroid syndrome in which patients display the features of CS and also brittle hair and nails [59,61]. The features of these NER-deficiency syndromes are well studied in animal models [62–66]. Importantly, studies with animal models have revealed that the severity of the progeroid features correlates well with the degree of DNA repair defects, suggesting causality [66,67].

Lastly, highly cytotoxic DSBs are primarily repaired either by the efficient but error-prone nonhomologous end-joining (NHEJ) pathway or the more precise homologous recombination (HR) pathway. NHEJ works in somatic cells (or proliferating cells in G1 stage) and is capable of joining the ends of the DNA strand via different sub-pathways, depending on the configuration of the DNA ends [68]; however, it works without a proper template, as it occurs independently of replication, and, therefore, often results in mutations (deletions or insertions). On the other hand, HR works in proliferating cells and is of particular importance during embryogenesis. After replication, HR uses the available identical copy of the damaged DNA to properly align the broken ends and repair the lesion [69], thus promoting cell survival without contributing to an increased mutagenic load. In addition, together with Fanconi's anaemia (FA) proteins, HR is also involved in ICL removal. In humans, mutations in key HR genes lead to a clear increase in cancer development, with mutations in the *BRCA1* and *BRCA2* genes being associated mostly, but not exclusively, with breast and ovarian cancer [70–72]. Mutations in HR genes can also lead to the development of FA/FA-related pathologies, characterized by bone marrow failure, developmental deficiencies and also an increased risk of cancer development [70].

The existence of such complex syndromes highlights the intricate relationship between DNA damage, ageing and cancer predisposition. Mutations in repair pathways that deal with mutagenic lesions often lead to cancer development while mutations in systems dealing with cytotoxic lesions (i.e. arrested transcription) are detrimental for normal growth and tissue homeostasis and thus contribute to an 'artificially accelerated' ageing process. This phenotypical dichotomy emphasizes the trade-off between the decisions a cell needs to take when facing irreparable DNA damage: minimize malignancy risk or maintain tissue functionality.

Moreover, the broad range of, sometimes highly specific, pathological outcomes strongly suggests that those genome instability syndromes, and physiological ageing itself, cannot be explained simply by the cell-autonomous effects of DNA damage. If the cell-autonomous DNA damage-induced increase in mutagenesis/cell death were the sole agent at play here, one would expect to observe similar tissue-unspecific pathologies independently of the affected repair pathway. Instead, different types of DNA damage appear to affect tissues differently, suggesting that complex signalling pathways might influence the whole organismal phenotype by coordinating specific systemic responses to damage.

# 5. Non-cell-autonomous DNA damage responses

The direct cell-autonomous consequences of DNA damage are unlikely to be the sole cause of both the complex pathological outcomes observed in patients with genome instability syndromes and the broad range of age-related phenotypes. Under this paradigm, the following questions arise: (i) Are there non-cell-autonomous responses promoting tissue dysfunction following DNA damage? (ii) Alternatively, are there compensatory non-cell-autonomous responses aiming to re-establish tissue homeostasis? (iii) Can different types of damage elicit specific systemic responses?

Regarding (i), we discuss below the influence of cellular senescence in the ageing process. Accumulating evidence shows the contribution of cellular senescence to age-related tissue dysfunction, and ablation of senescent cells via different mechanisms has shown potential in ameliorating multiple age-related phenotypes and even increasing lifespan [73–76]. Additionally, because of their heterogeneous aberrant secretory profiles (the senescent-associated secretory phenotype, SASP) [77,78], senescent cells are indeed capable of coordinating distinct non-cell-autonomous responses able to disrupt tissue homeostasis. Moreover, the tight links between cellular senescence, inflammation and stem cell exhaustion reflect the entanglement between different hallmarks of ageing and how multiple physiological layers orchestrate the onset of age-related functional decline.

Finally, regarding questions (ii) and (iii), we discuss recent findings in *C. elegans* and in mammalian models exemplifying compensatory stress responses, involving trans-tissue communication, elicited following different types of damage in order to maintain tissue functionality.

royalsocietypublishing.org/journal/rsob   Open Biol. 9: 190168

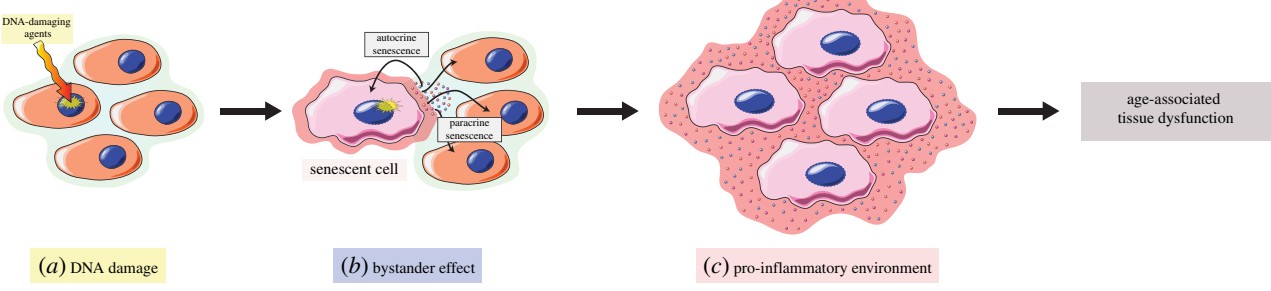

*(a) DNA damage*  *(b) bystander effect*  *(c) pro-inflammatory environment*

**Figure 2.** Non-cell-autonomous DNA damage responses contributing to age-associated tissue dysfunction. Cellular senescence can be elicited in response to a permanent DDR following exposure to DNA-damaging agents (*a*). Once established, senescent cells secrete a host of pro-inflammatory cytokines, chemokines, growth factors and matrix-remodelling enzymes (SASP), capable of coordinating distinct non-cell-autonomous responses. Via the SASP, senescent cells create a local pro-inflammatory environment that can reinforce their own senescent state (autocrine senescence) and, simultaneously, induce senescence in bystander cells (paracrine senescence) (*b*). This induction of senescence in bystander cells might be a relevant mechanism contributing to the reported age-associated accumulation of senescent cells in multiple tissues. Additionally, the resulting pro-inflammatory environment (*c*) might create a positive feedback loop, escalating the number of senescent cells within a tissue and the production of pro-inflammatory components, contributing to age-associated tissue dysfunction.

## 5.1. Cellular senescence, inflammation and ageing

Cellular senescence has been traditionally regarded as a state of irreversible cell-cycle arrest elicited by replicative exhaustion (replicative senescence) or in response to diverse, oncogenic or DNA-damaging stressors (oncogene-induced senescence and stress-induced premature senescence) [79–82]. Classical hallmarks of senescence include (i) promiscuous and highly heterogeneous gene expression; (ii) apoptosis resistance; and (iii) growth arrest [82]. For decades now, cells have been mainly identified as senescent by the presence of senescence-associated β-galactosidase (SA-βGal) activity [83]. Senescent cells display striking changes in gene expression when compared with their non-senescent counterparts [84–86]; major changes often include the over-expression of important cell-cycle inhibitors, including the cyclin-dependent kinase inhibitors p21 (CDKN1a/CIP1) and p16 (CDKN2a/INK4) [82,87–89], downstream effector proteins resulting from the activation of p53, thus linking the DDR to the establishment of cell-cycle arrest. In recent years, many other features have been associated with a senescent phenotype, including loss/redistribution of lamin B1 [90], accumulation of lipofuscin [91,92], loss of nuclear HMGB1 [93], telomere-associated DNA damage foci [94], senescence-associated heterochromatin foci [95] and senescence-associated mitochondrial dysfunction [96], to name but a few. Most striking, though, is the distinct secretome profile of senescent cells, termed SASP [97,98]. Senescent cells secrete a host of pro-inflammatory cytokines, chemokines, growth factors and matrix-remodelling enzymes capable of altering a tissue's microenvironment via autocrine and paracrine signalling, ultimately contributing to age-related tissue dysfunction [98,99]. Importantly, senescent cells, via the SASP and/or ROS production, have been shown to be capable of maintaining a state of chronic inflammation and inducing senescence in adjacent bystander cells, both *in vitro* and *in vivo* [100–104]. This induced senescence, via a 'bystander effect', might be a relevant mechanism leading to the reported age-dependent accumulation of senescent cells *in vivo* [105,106]. Even more worrisome, by creating a local chronic state of inflammation via the SASP, a small amount of senescent cells can convert other, otherwise healthy, cells into a senescent state, which in turn will escalate the production of pro-inflammatory components, creating a positive feedback loop potentially capable of affecting the organism in a systemic way (figure 2). This is especially concerning, as it has been reported that even relatively low numbers of senescent cells induce a bystander effect and can disrupt tissue homeostasis [75,104,107,108]; thus, senescent cells might play an active role as a driver of many age-related disorders and the process of physiological ageing itself.

The contribution of senescent cell accumulation to ageing is very well illustrated by the studies of Baker *et al.*, originally with BubR1 mice. BubR1 mice have severe genome instability due to defects in spindle assembly and, consequently, display multiple progeroid features and increased levels of senescent cells [73,109]. Notably, genetic clearance of p16-positive cells in the BubR1 background delayed both the onset and progression of age-related phenotypes [73]. Using the same system, clearance of naturally occurring p16-positive cells in a non-progeroid background increased both healthspan and lifespan [74]. These studies provide strong evidence that senescent cells are active drivers of the ageing process; nevertheless, it is important to note that senescent cells can be highly heterogeneous and not every senescent cell shows necessarily increased expression of p16, so it is possible that only a number of specific senescent cell populations (in this case, p16-expressing cells) are the actual drivers of the observed phenotypes.

Intriguingly, senescence-like phenotypes have been reported in postmitotic cells such as neurons [110], osteocytes [111], retinal cells [112], myofibres [104] and cardiomyocytes [113], among others, challenging the traditional view of senescence as a proliferation arrest-dependent programme.

Senescence is, however, not only tightly associated with physiological age-related phenotypes but also with multiple age-associated pathologies [114]. Cells with senescence-like phenotypes have been shown to accumulate in the lung in cases of idiopathic pulmonary fibrosis [115], in osteoarthritic joints [116] and in the liver in non-alcoholic fatty-liver disease [117], among other situations [114]. Correspondingly, transplantation of senescent cells can induce an osteoarthritic-like phenotype and impair function [75,116]. Conversely, removal of senescent cells with senolytic drug treatments has been shown to improve several healthspan parameters [75] and to improve tissue function in animal models of pulmonary fibrosis [115], atherosclerosis [118], hepatic steatosis [117] and obesity [119].

royalsocietypublishing.org/journal/rsob    Open Biol. 9: 190168

Senescence can be a highly heterogeneous phenotype, induced by multiple inputs and often leading to different and even opposite outputs [99]. In fact, senescent cells have been shown to promote both tumour suppression [80,120,121] and tumour progression [107,108,122,123], to contribute to wound repair [124] (but ultimately they may also drive the ageing process) and have been associated with numerous age-associated disorders [114]. Cellular senescence can thus be classified as an antagonistic hallmark of ageing [27] and a good representative of the antagonistic pleiotropy theory of ageing. Cellular senescence might have evolved as a tumour suppression mechanism with clearly beneficial effects early on; however, as senescent cell frequencies increase with age [105,106], the deleterious effects, mainly of the SASP, start to outweigh the initial, 'selected-through-evolution', beneficial effects.

The effects of the SASP are also inherently interconnected with other age-associated features, particularly the age-dependent increase in low-grade systemic inflammation (inflammageing) [125,126]. This increase in inflammation is not exclusively due to the SASP; enhanced activation of the nuclear factor kappa-light-chain-enhancer of activated B cells (NF-κB) transcription factor, the gradual inability of the immune system to remove sources of inflammation and autophagy dysfunction are also important contributors to age-associated inflammageing [127,128]. Overexpression of NF-κB is a known driving feature of ageing [129] and its inhibition delays cellular senescence and the onset of age-associated features [130]. Conversely, chronic activation of NF-κB induced by knockout of its nfkb1 subunit promotes cellular senescence by exacerbating telomere dysfunction and reduces the regenerative potential of tissues, thus accelerating ageing [100]. The reduction of tissues' regenerative potential is also one of the most striking features of ageing. This results mostly from functional exhaustion of stem cells in the majority of an organism's stem cell compartments [27], promoted not only by inflammation [100,131] but also by DNA damage [132] and cellular senescence [133,134], once again highlighting the entanglement of mechanisms and feedback loops driving the ageing process. Moreover, NF-κB activation, specifically in the hypothalamus, has been shown to inhibit the production of gonadotropin-releasing hormone, consequently contributing to the dysfunction of several other tissues [135]. Additionally, the age-associated decrease in hypothalamic stem cells and decrease in exosome secretion also accelerate the ageing speed and distal tissue dysfunction [135,136]. The hypothalamus thus appears to be involved in modulating systemic responses driving age-related pathology [137], reflecting the importance of the brain during ageing.

## 5.2. Compensatory stress responses, tissue functionality and longevity

Numerous studies from the past few decades have revealed that the ageing process is genetically regulated and cannot be explained simply as a consequence of damage accumulation. Pioneering studies with *C. elegans* have shown that lifespan can be significantly extended by mutations in single genes [138]. Currently, hundreds of genes have been identified in model organisms in which mutations lead to an increased lifespan—in some cases, up to a 10-fold increase [139]. Identification of these genes allowed the recognition of multiple so-called 'longevity-pathways', which, importantly, appear to be evolutionarily conserved in mammals. Among the best studied of these pathways are the above-mentioned CR and the insulin/insulin-like growth factor 1 (IGF-1) signalling (IIS) pathways.

The mechanisms accounting for the beneficial effects of CR remain somewhat elusive but their study is paramount in identifying cellular/systemic processes counteracting the age-associated increases in morbidity and mortality. Both the IIS and the target of rapamycin (TOR) pathways have been shown to mediate some of the beneficial effects of CR [23,140]. The inhibition of the TOR pathway, in particular, has been shown to have a very well-conserved role in mediating the CR-dependent lifespan extension among different organisms [141]. Reducing TOR activity leads, among other processes, to an increase in autophagy and a decrease in protein biosynthesis, both required for the CR-dependent increase in lifespan [23,141–143]. Nevertheless, the exact mechanisms by which these two processes exert their effects are still poorly understood.

Curiously, classical studies have shown that mice under CR are also more resistant to different acute stressors, including damage by surgical procedures, toxic drug administration and acute increase in ambient temperature [144], highlighting the intrinsic link between somatic maintenance and the retardation of the ageing process. In particular, studies in both humans and other animal models have shown that CR ameliorates oxidative damage, in particular to DNA and RNA [145,146], improves cellular quality control by promoting autophagy [147], promotes mitochondrial biogenesis [148] and impairs the SASP of senescent cells [149]. These and other mechanisms may very well be the drivers of the observed CR-associated increase in lifespan.

Regarding the IIS pathway, in mammals, the production of IGF-1 is promoted by the growth hormone (GH) produced and secreted from the pituitary gland. Dampening insulin signalling by manipulating components of this pathway (i.e. GH, the insulin/IGF-1 receptors or downstream effectors, such as FOXO) has been associated with an increase in longevity in both animal models and humans [150–152]. In *C. elegans*, dampening of the IIS pathway via mutations in the *daf-2* and *age-1* genes (the genes encoding the worm's orthologues of the insulin/IGF receptor and phosphatidylinositol 3-kinase, respectively) results in lifespan extension [138,139,150]. This effect is, however, dependent on the DAF-16 (the FOXO orthologue) transcription factor [138,150], the main IIS effector.

Importantly, the IIS pathway responds to DNA damage [153,154]. DAF-16 has been shown to be activated in response to persistent DNA damage in somatic tissues during the development of *C. elegans* in order to promote tissue functionality and allow growth to proceed; however, DAF-16 responsiveness to DNA damage is severely blunted with ageing [153]. Similarly, a recent proteome analysis of NER-deficient worms following UV exposure identified DAF-2 as a central hub, connecting different signalling nodes and coordinating a systemic response to permanent DNA damage [154]. Notably, the observed proteome changes resembled the ones naturally occurring during ageing [155,156], again underscoring the role of DNA damage accumulation during physiological ageing and hinting towards a systemic adaptive 'survival response' aiming to maintain tissue functionality

following damage. Additionally, multiple transcriptome analysis of NER-deficient mice also showed a dampening of the somatotropic axis in response to DNA damage [157–160], emphasizing that the same physiological mechanisms are shared between short- and long-lived models [159,161]. Of note, this same response was also observed in wild-type animals exposed to DNA-damaging agents [157]. These findings raise the intriguing possibility that low levels of, possibly localized, genotoxic stress are capable of triggering systemic stress responses and actually contribute to the maintenance of tissue functionality. Supporting this hypothesis, NER deficiency in *C. elegans* has been shown to increase the expression of 'stress-responsive' genes and further increase the lifespan of the already long-lived *daf-2* mutants [162].

Systemic responses to tissue-specific DNA damage have previously been observed in both *Drosophila melanogaster* and *C. elegans* [163,164]. In *D. melanogaster* larvae, DNA damage in the epidermis triggers an immune response dependent on c-Jun N-terminal kinase (JNK) and Janus kinase (JAK)/ signal transducer and activator of transcription (STAT) signalling, consequently limiting insulin-like peptide secretion by the central nervous system and activating FOXO [163]. JNK signalling is a prominent stress-responsive pathway in *D. melanogaster* and JAK/STAT signalling has been shown multiple times to be involved in systemic responses promoting tissue regeneration in injury models [165,166].

In *C. elegans*, the germline has been known for many years to be able to influence somatic maintenance, with germ cell-deficient worms being long-lived and stress-resistant [167,168]. In addition, somatic tissues are able to respond to damage in germ cells. GG-NER mutant worms, unable to remove DNA lesions in the germline, have surprisingly resistant somatic tissues. This stress response is mediated by an innate immune response in the germline triggered by the mitogen-activated protein kinase 1 (MPK-1, the extracellular signal-regulated kinases 1/2 (ERK1/2) MAPK homologue), which later becomes systemically established. This coordinated mechanism has been termed 'germline DNA damage-induced stress resistance' [164] and appears to be a mechanism evoked in order to extend an organism's lifespan following localized DNA damage. This provides the organism with more time to repair the damage present in the germline and prevent the transmission of harmful mutations to the next generation without sacrificing too much of the amount of progeny produced [169]. Supporting this explanation, offspring production is transiently reduced following exposure to DNA-damaging agents, but resumes after the period used for DNA repair and lasts past the normal reproductive period of non-damaged worms [164].

Lastly, it is important to mention the role of the neuronal system in the coordination of systemic stress responses. Neurons are ideally equipped for this as they are capable of (i) secreting chemicals able to reach distal tissues; (ii) sensing and processing environmental cues; and (iii) integrating those signals and coordinating physiological responses accordingly. Many reports have demonstrated the crucial role of the neuronal system in regulating longevity and proteostasis in *C. elegans*, mostly via communication with the intestine, presumably via neuroendocrine signals [170–178]. In the future, it would be of importance to better characterize the types of neuronal responses elicited following distinct DNA lesions and investigate possible neuron-mediated distal stress responses following DNA damage.

# 6. Concluding remarks

Research over past decades has elucidated the role of genomic instability as a root cause of ageing. The observed age-dependent accumulation of somatic mutations in the genome [26] and the accelerated ageing phenotypes caused by deficiencies in DNA repair systems provide compelling evidence supporting an active role for intrinsic DNA damage in mediating loss of tissue functionality with ageing. Still, the broad range of phenotypic variability within ageing populations strongly suggests that complex signalling pathways might coordinate specific systemic responses to DNA damage. These systemic responses have become increasingly apparent in multiple species and appear to have a major role not only during the physiological ageing process but also in response to acute stress. Importantly, these responses represent perfect examples of the intricate connection between DNA damage and other hallmarks of ageing, such as cellular senescence, stem cell exhaustion and altered intercellular communication [27], which can all occur as a consequence of the DDR. Nevertheless, the interplay between cell-autonomous and these non-cell-autonomous responses is still somewhat poorly understood. Future studies should aim to better understand how different types of DNA lesions can elicit such phenotypic variability by identifying key effector systemic signalling networks. For this, *C. elegans* might prove especially useful, as the nematode worm is an ideal model organism to study the consequences of damage in fully differentiated, postmitotic tissues. This system will facilitate the identification of mechanisms orchestrating the systemic consequences of DNA damage and will surely provide important novel insights about the impact of genome instability in physiological ageing and age-related pathology.

Data accessibility. This article has no additional data.

Authors' contributions. P.F.L.d.S. and B.S. wrote the article.

Competing interests. We declare we have no competing interests.

Funding. P.F.L.d.S received support from the Cologne Graduate School of Ageing Research. B.S. acknowledges funding from the Deutsche Forschungsgemeinschaft (grant nos. SCHU 2494/3-1, SCHU 2494/ 7-1, CECAD, SFB 829, KFO 286, KFO 329 and GRK2407), the Deutsche Krebshilfe (grant no. 70112899) and COST Action (grant no. BM1408).

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
