## [Reviewer comments · Open Biology]

Review History

RSOB-19-0168.R0 (Original submission)

Review form: Reviewer 1

Recommendation

Accept as is

Do you have any ethical concerns with this paper?

No

Comments to the Author

In a thought-provoking review article, da Silva and Schumacher elegantly present and judiciously discuss recent advances and unanswered questions in understanding mechanisms that regulate cellular and organismal aging in response to an age-related accumulation of DNA damage. The authors critically analyze the recent progress in appreciating the complexity of systemic signaling networks that define the pace of the aging process by integrating various cell-autonomous and cell-nonautonomous DNA damage responses. They hypothesize that different types of the intrinsic DNA damage may differently affect these signaling networks of longevity regulation in postmitotic tissues of the nematode worm *Caenorhabditis elegans*.

This well organized and clearly written manuscript is a must-read for anyone interested in applying advanced intellectual and mechanistic approaches to studying aging and developing novel therapeutic strategies for the deceleration of the aging process. As such, the manuscript provides a significant new contribution to the field. Therefore, it is suitable for Open Biology, and I recommend accepting it for publication.

Review form: Reviewer 2

Recommendation

Accept with minor revision (please list in comments)

Do you have any ethical concerns with this paper?

No

Comments to the Author

DNA damage response in aging
Da Silva, Paulo and Schumacher, Bjorn
Open Biology 2019.

In this review, da Silva and Schumacher discuss multiple components of aging at the molecular, cellular, environmental and organismal levels. Schumacher has an extensive publication record of original research and review articles addressing cellular and molecular mechanisms of aging, particularly DNA damage, in *C. elegans*. Da Silva is first author on a recently published article in *Aging Cell* identifying the spread of senescence among cells in vivo via the “bystander effect.” Consequently, particular attention is given to DNA damage and cellular senescence, as illustrated in the two accompanying Figures.

The topic of aging is always timely, and the ideas discussed in this review offer novel insights into how external and internal events shape aging and longevity in vivo. The authors extend findings in *C. elegans* to research directions in preclinical and clinical studies, creating a connection between basic science and translational relevance. The manuscript is very well written but could use one more pass to catch typographical errors.

The Introduction addresses general concepts and theories of aging and senescence, and touches on the interplay between environment, genetics and organismal compromises between reproduction and longevity. Two or three summary sentences at the end of the Introduction section would serve to tie these “big picture” concepts to those that follow, and to state the overall trajectory, themes and goals of the review.

Future research directions are discussed with an emphasis on *C. elegans* as a model organism for studying DNA damage and aging in postmitotic tissues.

The graphics are nicely rendered, suggestions for improving the Figures are as follows:

Figure 1: Numbering the text boxes to correspond with the red boxes on the graphic will facilitate visual interpretation of the figure, i.e. Mismatch repair (MMR) is 1, Single strand break repair (SSBR) is 2, etc. Similarly, the NHEJ and HR text boxes can both be labeled 5 but could be flipped to reflect the order in which they are mentioned in the Figure legend.

Figure 2: Assign each part of the figure a letter (i.e. left-to-right A, B, C, D). Include summary labels for A and B similar to the one for “pro-inflammatory environment” under C, for example

“DNA damage” under A and “bystander effect” under B. Update the figure legend to incorporate changes in the graphic.

Figure 3?: If allowed, an overview figure depicting relationships among the molecular, cellular, signaling, organismal and even environmental aspects of aging would help tie together the concepts discussed in the review.

Decision letter (RSOB-19-0168.R0)

21-Oct-2019

Dear Dr Schumacher

We are pleased to inform you that your manuscript RSOB-19-0168 entitled "DNA damage responses in aging" has been accepted by the Editor for publication in Open Biology. The reviewer(s) have recommended publication, but also suggest some minor revisions to your manuscript. Therefore, we invite you to respond to the reviewer(s)' comments and revise your manuscript.

Please submit the revised version of your manuscript within 7 days. If you do not think you will be able to meet this date please let us know and we can extend this deadline for you.

- 1) A text file of the manuscript (doc, txt, rtf or tex), including the references, tables (including captions) and figure captions. Please remove any tracked changes from the text before submission. PDF files are not an accepted format for the "Main Document".
- 2) A separate electronic file of each figure (tiff, EPS or print-quality PDF preferred). The format should be produced directly from original creation package, or original software format. Please note that PowerPoint files are not accepted.
- 3) Electronic supplementary material: this should be contained in a separate file from the main text and meet our ESM criteria (see <http://royalsocietypublishing.org/instructions->

authors#question5). All supplementary materials accompanying an accepted article will be treated as in their final form. They will be published alongside the paper on the journal website and posted on the online figshare repository. Files on figshare will be made available approximately one week before the accompanying article so that the supplementary material can be attributed a unique DOI.

Online supplementary material will also carry the title and description provided during submission, so please ensure these are accurate and informative. Note that the Royal Society will not edit or typeset supplementary material and it will be hosted as provided. Please ensure that the supplementary material includes the paper details (authors, title, journal name, article DOI). Your article DOI will be 10.1098/rsob.2016[last 4 digits of e.g. 10.1098/rsob.20160049].

4) A media summary: a short non-technical summary (up to 100 words) of the key findings/importance of your manuscript. Please try to write in simple English, avoid jargon, explain the importance of the topic, outline the main implications and describe why this topic is newsworthy.

Images

Data-Sharing

It is a condition of publication that data supporting your paper are made available. Data should be made available either in the electronic supplementary material or through an appropriate repository. Details of how to access data should be included in your paper. Please see <http://royalsocietypublishing.org/site/authors/policy.xhtml#question6> for more details.

Data accessibility section

Sincerely,

The Open Biology Team

<mailto:openbiology@royalsociety.org>

Reviewer(s)' Comments to Author:

Referee: 1

Comments to the Author(s)

In a thought-provoking review article, da Silva and Schumacher elegantly present and judiciously discuss recent advances and unanswered questions in understanding mechanisms that regulate cellular and organismal aging in response to an age-related accumulation of DNA damage. The authors critically analyze the recent progress in appreciating the complexity of

systemic signaling networks that define the pace of the aging process by integrating various cell-autonomous and cell-nonautonomous DNA damage responses. They hypothesize that different types of the intrinsic DNA damage may differently affect these signaling networks of longevity regulation in postmitotic tissues of the nematode worm *Caenorhabditis elegans*.

This well organized and clearly written manuscript is a must-read for anyone interested in applying advanced intellectual and mechanistic approaches to studying aging and developing novel therapeutic strategies for the deceleration of the aging process. As such, the manuscript provides a significant new contribution to the field. Therefore, it is suitable for Open Biology, and I recommend accepting it for publication.

Referee: 2

Comments to the Author(s)

In this review, da Silva and Schumacher discuss multiple components of aging at the molecular, cellular, environmental and organismal levels. Schumacher has an extensive publication record of original research and review articles addressing cellular and molecular mechanisms of aging, particularly DNA damage, in *C. elegans*. Da Silva is first author on a recently published article in *Aging Cell* identifying the spread of senescence among cells in vivo via the “bystander effect.” Consequently, particular attention is given to DNA damage and cellular senescence, as illustrated in the two accompanying Figures.

The topic of aging is always timely, and the ideas discussed in this review offer novel insights into how external and internal events shape aging and longevity in vivo. The authors extend findings in *C. elegans* to research directions in preclinical and clinical studies, creating a connection between basic science and translational relevance. The manuscript is very well written but could use one more pass to catch typographical errors.

The Introduction addresses general concepts and theories of aging and senescence, and touches on the interplay between environment, genetics and organismal compromises between reproduction and longevity. Two or three summary sentences at the end of the Introduction section would serve to tie these “big picture” concepts to those that follow, and to state the overall trajectory, themes and goals of the review.

Future research directions are discussed with an emphasis on *C. elegans* as a model organism for studying DNA damage and aging in postmitotic tissues.

The graphics are nicely rendered, suggestions for improving the Figures are as follows:

Figure 1: Numbering the text boxes to correspond with the red boxes on the graphic will facilitate visual interpretation of the figure, i.e. Mismatch repair (MMR) is 1, Single strand break repair (SSBR) is 2, etc. Similarly, the NHEJ and HR text boxes can both be labeled 5 but could be flipped to reflect the order in which they are mentioned in the Figure legend.

Figure 2: Assign each part of the figure a letter (i.e. left-to-right A, B, C, D). Include summary labels for A and B similar to the one for “pro-inflammatory environment” under C, for example “DNA damage” under A and “bystander effect” under B. Update the figure legend to incorporate changes in the graphic.

Figure 3?: If allowed, an overview figure depicting relationships among the molecular, cellular, signaling, organismal and even environmental aspects of aging would help tie together the concepts discussed in the review.

Author's Response to Decision Letter for (RSOB-19-0168.R0)

See Appendix A.

Decision letter (RSOB-19-0168.R1)

28-Oct-2019

Dear Professor Schumacher

We are pleased to inform you that your manuscript entitled "DNA damage responses in aging" has been accepted by the Editor for publication in Open Biology.

Sincerely,

The Open Biology Team
mailto: openbiology@royalsociety.org

Appendix A

Response to Reviewers

We would like to thank the reviewers for their enthusiasm for our review article. We have amended the text according to the reviewers' suggestions.

Referee: 1

Comments to the Author(s)

In a thought-provoking review article, da Silva and Schumacher elegantly present and judiciously discuss recent advances and unanswered questions in understanding mechanisms that regulate cellular and organismal aging in response to an age-related accumulation of DNA damage. The authors critically analyze the recent progress in appreciating the complexity of systemic signaling networks that define the pace of the aging process by integrating various cell-autonomous and cell-nonautonomous DNA damage responses. They hypothesize that different types of the intrinsic DNA damage may differently affect these signaling networks of longevity regulation in postmitotic tissues of the nematode worm *Caenorhabditis elegans*.

This well organized and clearly written manuscript is a must-read for anyone interested in applying advanced intellectual and mechanistic approaches to studying aging and developing novel therapeutic strategies for the deceleration of the aging process. As such, the manuscript provides a significant new contribution to the field. Therefore, it is suitable for Open Biology, and I recommend accepting it for publication.

We very much appreciate the reviewer's recognition of the importance of our article.

Referee: 2

Comments to the Author(s)

In this review, da Silva and Schumacher discuss multiple components of aging at the molecular, cellular, environmental and organismal levels. Schumacher has an extensive publication record of original research and review articles addressing cellular and molecular mechanisms of aging, particularly DNA damage, in *C. elegans*. Da Silva is first author on a recently published article in *Aging Cell* identifying the spread of senescence among cells in vivo via the "bystander effect." Consequently, particular attention is given to DNA damage and cellular senescence, as illustrated in the two accompanying Figures.

The topic of aging is always timely, and the ideas discussed in this review offer novel insights into how external and internal events shape aging and longevity in vivo. The authors extend findings in *C. elegans* to research directions in preclinical and clinical studies, creating a connection between basic science and translational relevance. The manuscript is very well written but could use one more pass to catch typographical errors.

The Introduction addresses general concepts and theories of aging and senescence, and touches on the interplay between environment, genetics and organismal compromises between reproduction and longevity. Two or three summary sentences at the end of the Introduction section would serve to tie these "big picture" concepts to those that follow,

and to state the overall trajectory, themes and goals of the review.

Future research directions are discussed with an emphasis on *C. elegans* as a model organism for studying DNA damage and aging in postmitotic tissues.

We would like to thank the reviewer for the interest in our article and the helpful suggestions. We have now added a new paragraph to the introduction to more easily present the overall picture to the reader.

The graphics are nicely rendered, suggestions for improving the Figures are as follows:

Figure 1: Numbering the text boxes to correspond with the red boxes on the graphic will facilitate visual interpretation of the figure, i.e. Mismatch repair (MMR) is 1, Single strand break repair (SSBR) is 2, etc. Similarly, the NHEJ and HR text boxes can both be labeled 5 but could be flipped to reflect the order in which they are mentioned in the Figure legend.

Thank you for this suggestion. We have added numbering.

Figure 2: Assign each part of the figure a letter (i.e. left-to-right A, B, C, D). Include summary labels for A and B similar to the one for “pro-inflammatory environment” under C, for example “DNA damage” under A and “bystander effect” under B. Update the figure legend to incorporate changes in the graphic.

Thank you for this suggestion. We have now added letter to make the flow of the figure more accessible.

Figure 3?: If allowed, an overview figure depicting relationships among the molecular, cellular, signaling, organismal and even environmental aspects of aging would help tie together the concepts discussed in the review.

We believe that the current figures sufficiently illustrate the main contents of the article.